# Identification of Electrical Tree Aging State in Epoxy Resin Using Partial Discharge Waveforms Compared to Traditional Analysis

**DOI:** 10.3390/polym15112461

**Published:** 2023-05-26

**Authors:** Roger Schurch, Osvaldo Munoz, Jorge Ardila-Rey, Pablo Donoso, Vidyadhar Peesapati

**Affiliations:** 1Department of Electrical Engineering, Universidad Tecnica Federico Santa Maria, Valparaiso 2390123, Chile; jorge.ardila@usm.cl; 2Department of Engineering and Design, Chilean Independent System Operator, Santiago 9020000, Chile; osvaldo.munoz@coordinador.cl; 3Department of Electrical and Electronic Engineering, The University of Manchester, Manchester M13 9PL, UK; pablo.donosodaille@manchester.ac.uk (P.D.);

**Keywords:** electrical trees, partial discharges, epoxy resin, pulse waveform

## Abstract

Electrical treeing is one of the main degradation mechanisms in high-voltage polymeric insulation. Epoxy resin is used as insulating material in power equipment such as rotating machines, power transformers, gas-insulated switchgears, and insulators, among others. Electrical trees grow under the effect of partial discharges (PDs) that progressively degrade the polymer until the tree crosses the bulk insulation, then causing the failure of power equipment and the outage of the energy supply. This work studies electrical trees in epoxy resin through different PD analysis techniques, evaluating and comparing their ability to identify tree bulk-insulation crossing, the precursor of failure. Two PD measurement systems were used simultaneously—one to capture the sequence of PD pulses and another to acquire PD pulse waveforms—and four PD analysis techniques were deployed. Phase-resolved PD (PRPD) and pulse sequence analysis (PSA) identified tree crossing; however, they were more sensible to the AC excitation voltage amplitude and frequency. Nonlinear time series analysis (NLTSA) characteristics were evaluated through the correlation dimension, showing a reduction from pre- to post-crossing, and thus representing a change to a less complex dynamical system. The PD pulse waveform parameters had the best performance; they could identify tree crossing in epoxy resin material independently of the applied AC voltage amplitude and frequency, making them more robust for a broader range of situations, and thus, they can be exploited as a diagnostic tool for the asset management of high-voltage polymeric insulation.

## 1. Introduction

Electrical energy is generated, transmitted, and distributed through power system networks, and the utilization of that energy is through many electrical apparatuses. One important aspect affecting the reliability of power systems and apparatuses is the dielectric (equipment insulation) reliability. Insulation systems are key elements in the safe and reliable operation of not only high-voltage equipment, but also power electronic devices that work at high electric fields, stressing dielectric materials [1]. Epoxy resin is one important insulating material within solid polymeric insulation, with many practical applications in electrical engineering due to its good mechanical and chemical properties, heat resistance and thermal stability, electrical insulation, and ease of manufacturing [2,3,4]. Epoxy resin is an amorphous polymeric material resulting from a thermoset reaction between a resin and a hardener. The most common epoxy used is the diglycidyl ether of bisphenol A (DGEBA), which is the result of a reaction between bisphenol A and epichlorohydrin [5]. The use of epoxy resin in electrical engineering includes transformers, electrical machines, insulators, bushings, switchgear, power electronic devices, and circuit boards, among other electrical equipment [6].

The main mechanism of degradation and long-term failure in solid polymeric insulation (including epoxy resin material) is the formation of electrical trees [7,8]. They are hollow tubular channels of degradation with 1–10 µm diameters that resemble botanical trees; see Figure 1. Electrical trees grow under high electrical stress and are associated with partial discharge (PD) activity, which is the localized dielectric breakdown of a small portion of an insulation system. Once the electrical tree crosses the bulk insulation, the failure is imminent but not necessarily soon after the crossing. PDs are used to diagnose the insulation state of high-voltage equipment. However, the presence of harmonics due to the increased use of power electronics devices needed for low-carbon power networks creates difficulties for PD diagnosis due a great diversity of superimposed frequencies, complicating the interpretation and analysis of PD via conventional methods [9]. Most studies on electrical trees have analyzed the case of pure industrial frequency, but recent findings have shown that the resulting structure and time-to-breakdown of electrical trees grown in epoxy resin are influenced by the voltage excitation waveform [10].

PDs in electrical trees have been analyzed using more traditional techniques such as phase-resolved partial discharge (PRPD) [11] and pulse sequence analysis (PSA) [12], and also other techniques such as nonlinear time series analysis (NLTSA) [13] and pulse waveform analysis [14]. Using information from PD waveforms, electrical trees have been characterized by identifying their stage of development [15,16]. Even though these techniques have been used to describe electrical tree progression, they have not been compared in their ability to identify tree crossing, which is the most critical event in tree development and is a precursor to final insulation breakdown. The aim of this paper is to identify electrical tree crossing in epoxy resin insulation using PRPD, PSA, NLTSA, and PD waveform analysis techniques and compare them in their ability to carry out this task.

## 2. Materials and Methods

Samples were prepared in epoxy resin Mepox 1124(cl)/2262 (DGEBA type of transparent resin), with the resin being manufactured by Kumho P&B Chemicals and the hardener by Yunteh Industries. The samples’ geometry was conventional needle-plane configuration with a separation of 2 ± 0.3 mm between the energized metallic needle and the bottom grounded-plane, as shown in Figure 2a. The needle was placed in the sample during the pot life of the resin, after which, it was mixed with the hardener, degassed, and poured into the mold; thus, the epoxy resin cured with the needle already inserted. This epoxy was a cold curing epoxy system, with a mixing ratio of 2:1 in weight of epoxy/hardener. For the resin curing and post-curing process, the temperature was set at 25 °C for 24 h and then at 50 °C for 15 h. This sample preparation procedure facilitated the similarity of the samples and the repeatability of the results. The initiation of trees in the specimens was carried out through AC voltage–frequency steps applied progressively, starting at 10 kV (RMS)-50 Hz until reaching 14 kV-550 Hz, taking 5 min per step, as explained in [15].

For the growth of electrical trees in the specimens, the experiments were carried out under monitored laboratory conditions using the test circuit shown in Figure 2b. A high-voltage amplifier (HVA) Trek model 20/20C-HS (up to 20 kVpeak) was used as a high voltage source, which amplifies the signal coming from a signal generator (SG). The PD measurement circuit was the balanced circuit with a subtracting circuit (SC), presented in the IEC 60270 standard [17]. PDs that originated in the treeing sample (C_a_) were measured by simultaneously using two PD measuring systems: the commercial equipment Omicron MPD600 and the combination of an oscilloscope (Keysight Infiniium S DSOS104A) and a High-Frequency Current Transformer (HFCT, Techimp—bandwidth of 80 MHz). The image monitoring and recording of tree progression were carried out with a Canon T6 reflex camera with a Canon MP-E-65 mm f/2.8 1–5× macro lens. For more details on the measurement system, see [15].

As shown in Table 1, eight tests at different voltage and frequency levels were carried out. Two AC RMS voltage levels for each frequency (50, 150, 350, and 450 Hz) were chosen. For the case of Sample 350-B, the voltage level was 12 kV instead of 14 kV as for the other frequencies. This was because in the data from the experiment with 14 kV and 350 Hz, the intervals of interest were not recorded since the oscilloscope was in its dead time due to an internal saving process of the fast memory scheme.

As the data analyzed for this paper were obtained from two measurement systems, MPD 600 and an oscilloscope, different procedures were applied in each case. The analysis intervals from both measurements were the same, and their selection was dependent on the oscilloscope measurements, which were not continuous recordings as for Omicron. The data that came from the Omicron measuring system contained, for each PD pulse, the phase angle, time of occurrence, and the apparent charge to construct PRDP and PSA patterns. The data were filtered to reduce the presence of noise in the measurement; this was achieved by discarding all the discharges with apparent charges below a threshold defined by a test, which in this case varied from 3 to 5 pC. The other PD measuring system was the oscilloscope. A signal source separation process was applied to the measured waveform data to eliminate signals associated with noise, following the procedure described in [18]. This separation technique is based on the calculation of two parameters associated with the temporal and spectral behavior of PD signals. The first parameter is the skewness (S), which statistically characterizes the asymetry of the absolute value of the PD amplitude (charge). The second parameter is the ‘weighted maximum frequency’ (F_MW_), which characterizes the frequency spectrum of PD signals where their energy content is more relevant. With these two parameters, a 2D graph (map) is plotted [18]. The result is a map with two clearly identified clusters: one associated with PDs coming from electrical trees, and other associated with noise and perturbances. PDs belonging to the cluster associated with noise are then eliminated, and only PDs resulting from electrical tree growth are analyzed.

To visualize the evolution of the parameters, graphs such as the one in Figure 3 were generated. At certain intervals (bands in red), the average of the parameters was calculated and plotted together with the length of the tree at the same instant of time to visualize trends. The selected intervals were taken with the following criteria: duration of time between 0.5 and 2 min, over 1000 PD pulses measured, and a minimum of three selected intervals in the complete development of the tree.

## 3. Analysis Techniques

### 3.1. Phase-Resolved Partial Discharge

Among the main advantages of using the PRPD analysis technique is the extensive amount of previous research and accumulated experience, as well as the relative ease of calculation it represents. However, it also has disadvantages, such as availability problems relating to pure sinusoidal HV sources for PD tests (diagram distortion with sources with harmonic contamination) or measurement problems in noise-contaminated environments. In PRPD analysis, each PD event is represented by its phase of occurrence and charge amplitude. PD information is measured during several cycles of the external voltage and then graphed in a 2D plot, usually known as PRPD diagram or ‘pattern’, such as the one shown in Figure 4. This method does not require the acquisition of high-frequency signals. Therefore, the technological requirements are less demanding than other methods, facilitating the diffusion of PRPD in industry and academia.

The PRPD analysis used here has two approaches: qualitative and quantitative. The qualitative approach uses the visualization of PRPD patterns, using the applied voltage as a reference signal and plotting the phase of occurrence of the pulses and their apparent charges. This representation allows the creation of patterns associated with different PD sources and, therefore, can be used to identify PD sources. However, when using just PRPD patterns, the classification is not easy, and it usually requires highly trained human experts [19]. On the other hand, the quantitative approach analyzes statistical parameters obtained from data associated with PRPD analysis. In this paper, three statistical distributions derived from PRPD data were considered: histograms of the phase of occurrence in positive (C(+)) and negative (C(−)) cycles and a histogram of the discharge amplitude. Skewness, kurtosis, mean value, and variance were calculated for each distribution, obtaining 12 parameters for each PRPD diagram. The dissipated power parameter ($P_{dis}$), repetition rate (n, #PDs/min), and 95 percentile charge ($Q_{95}$) were also calculated for each analyzed stage. $P_{dis}$ and n are defined as follows:(1)$$P_{dis}=\frac{1}{\Delta t}\sum^{\;}q_{i}U_{i}$$
(2)$$n=\frac{N}{\Delta t}$$
where $q_{i}$ is the discharge amplitude, $U_{i}$ is the voltage associated with the i-th PD event, N is the total number of discharges, and Δt is the observation time interval in seconds. For details regarding parameters’ calculations, consult [17,20].

### 3.2. Pulse Sequence Analysis

Partial discharge in solid dielectrics relates not only to the externally applied voltage but also to the local electric field in the cavity; however, it is not easy to measure the local electric field which produces the discharge. Thus, pulse sequence analysis (PSA) was developed to obtain information about the changes in the local electric field from variables that could be measured externally. To represent each PD pulse, PSA could use the data of the charge amplitude $q_{n}$, the time of occurrence $t_{n}$, the voltage of occurrence $U_{n}$, or the phase of occurrence $\phi_{n}$. The information regarding a PD event parameter is plotted recursively, generally in a 2D plot [12,21].

PSA focuses on a qualitative approach to the study of PD patterns for which the visualization of graphs is essential. This tool considers the relationship of each pulse with its neighbors to avoid the loss of information on the physical phenomenon associated with the variation in the electric field. PSA proposes various patterns to analyze, but in the case of this work, the main analyzed pattern was $\Delta U_{\left(n-1\right)}/\Delta t_{\left(n-1\right)}$ v/s $\Delta U_{n}/\Delta t_{n}$, which, due to its sensitivity, has previously been used to describe electrical arborization [22,23].

In order to simplify the comparison between tests at different voltage and frequency levels, the patterns were normalized utilizing Equation (1), obtained from [22].
(3)$$\frac{\Delta U_{n}}{\Delta t_{n}}=\frac{U_{n}-U_{\left(n-1\right)}}{t_{n}-t_{\left(n-1\right)}}\cdot \frac{1}{V_{b}\cdot f_{b}}$$
where $V_{b}$ is the RMS test voltage and $f_{b}$ is the test frequency. For more information regarding the calculation and visualization methodology, consult [22].

### 3.3. Nonlinear Time Series Analysis

NLTSA is a set of tools derived from nonlinear dynamic system theory usually related to mathematic chaos. This paper used the correlation dimension $D_{2}$ to characterize electrical tree stages using the methodology deployed in [22]. The algorithm to calculate this parameter is based on Takens’ embedding method, which allows for the reconstruction of an approximation of the system’s state space from the measurement of a single observable variable. To explain this, let us assume we make *N* measurements of a single variable *x*; therefore, we have a time series $X=\left(x_{1},\;\ldots ,\;x_{N}\right)$. The idea of the reconstruction method is to consider a set of vectors
(4)$$Y_{k}=\left(x_{k},\;x_{k+T},\;x_{k+2T},\;\ldots ,\;x_{k+\left(m-1\right)T}\right)\;$$
of length m (also known as embedding dimension), constructed from the series X, where a time shift T>0 is allowed (this is also called delay embedding). Provided m is large enough, and T is chosen appropriately, the set of vectors given by $Y_{k}$ can be considered as an approximation of the space state trajectory of the actual dynamic system [24].

In this context, the correlation dimension can be interpreted as an estimation of the fractal dimension of the state space trajectory of the reconstructed dynamical systems. Furthermore, provided some formal requirements are met, Takens’ method ensures that the reconstructed space’s correlation dimension is approximately equal to that in the actual state space. This is key in practical applications of correlation dimension, such as PD analysis, because it allows for the characterization of PD dynamics through correlation dimension $D_{2}$. To calculate $D_{2}$, it is necessary to calculate the correlation sum using the following equation:(5)$$C_{2}\left(\epsilon \right)=\frac{2}{N\left(N-1\right)}\overset{N}{\underset{i=1}{\sum }}\overset{N}{\underset{j=i+1}{\sum }}\Theta \left(\epsilon -\|Y_{i}-Y_{j}\|\right)$$

In this equation, *N* is the total number of PD, $Y_{i}$ is a reconstructed vector, and Θ() is the Heaviside function. Finally, $D_{2}$ is estimated by finding the range ϵ where:(6)$$\ln C\left(\epsilon \right)\approx D_{2}\cdot \ln\epsilon +constant$$

Applying NLTSA tools is of interest in electrical treeing since this phenomenon can be analyzed as a nonlinear system [25]. In this context, the time series analyzed are the PD measurements [22]; thus, PDs measured with the MPD 600 were used to calculate the correlation dimension in this paper. The workflow followed to calculate the correlation dimension is described in more detail in [22]. The embedding delay T was calculated using the algorithm described in [26]. This paper used MATLAB implementation to calculate the correlation dimension $D_{2}$ for each sample at its pre-crossing and post-crossing stages. 

### 3.4. Partial Discharge Waveform Parameters

This method of discharge analysis is based on characterizing the waveform of each pulse by means of parameters in the time or frequency domain which are taken to graphic schemes to search for patterns or relationships that provide relevant information. These have been mainly used for partial discharge classification and noise filtering. Among these techniques, we can find the time frequency maps [27] and the power ratio maps [28]. In this work, six parameters that have shown their ability to identify tree development stages were used to contrast with the results of the PRPD and PSA patterns. These parameters were $A_{t}$, f, $\sigma_{T}$, $\sigma_{F}$, PRL, and PRH, which are briefly explained as follows. 

#### 3.4.1. Parameters $A_{t}$ and f

These parameters were proposed in [29] and selected among those with the best behavior for electrical tree development [15]. The parameters are defined by:(7)$$A_{t}=\int_{tp}^{tf}h_{pu}^{*}\left(t\right)dt$$
(8)$$h_{pu}^{*}\left(t\right)=\frac{h\left(t\right)}{I_{p}}\;\;\;\;\;$$
(9)$$f=\frac{\int_{f1}^{f2}f*F\left(f\right)df}{\int_{f1}^{f2}F\left(f\right)df}$$
where h(t) is the enveloping function of the pulse, which was calculated using Hilbert transform; $I_{p}$ is the peak value of h(t); $h_{pu}\left(t\right)$ is the normalized enveloping function; F(t) is the Fourier transform of the pulse; and f1 and f2 define the frequency interval where the amplitude of the spectrum is higher than 70% of its maximum.

#### 3.4.2. Parameters $\sigma_{T}$ and $\sigma_{F}$

The technique of time frequency maps [27] is described by the equivalent duration of the waveform ($\sigma_{T}$) parameter and the equivalent bandwidth ($\sigma_{F}$) of the spectrum for each PD pulse. The parameters are obtained from the following equations:(10)$$\tilde{s}\left(t\right)=\frac{s\left(t\right)}{\sqrt{\int_{0}^{T}s{\left(t\right)}^{2}dt}}$$
(11)$$\sigma_{T}=\sqrt{\int_{0}^{T}{\left(t-t_{0}\right)}^{2}\tilde{s}{\left(t\right)}^{2}dt}$$
(12)$$\sigma_{F}=\sqrt{\int_{0}^{\infty }f^{2}{\left|\tilde{s}\left(f\right)\right|}^{2}df}$$
where $\tilde{s}\left(t\right)$ is the normalization of the PD signal acquired, f is the frequency, $\tilde{s}\left(f\right)$ is the Fourier transform of $\tilde{s}\left(t\right)$ and $t_{0}$ is the ‘temporal gravity center’ of the normalized signal, defined by:(13)$$t_{0}=\int_{0}^{T}t\;\tilde{s}{\left(t\right)}^{2}dt$$

#### 3.4.3. Parameters PRL and PRH

These parameters are derived from the calculation of the spectral power of the PD signals in two frequency bands [28] and then normalized by dividing them by the total power of the signal. Thus, the power ratio for low frequencies (PRL) and the power ratio for high frequencies (PRH) are defined by:(14)$$PRL=\frac{\sum_{f_{1L}}^{f_{2L}}{\left|s\left(f\right)\right|}^{2}}{\sum_{0}^{f_{t}}{\left|s\left(f\right)\right|}^{2}}$$
(15)$$PRH=\frac{\sum_{f_{1H}}^{f_{2H}}{\left|s\left(f\right)\right|}^{2}}{\sum_{0}^{f_{t}}{\left|s\left(f\right)\right|}^{2}}$$
where s(f) is the magnitude of the Fourier transform of the pulse signal s(t), and $f_{t}$ is the maximum evaluated frequency. The frequency bands can be chosen according to the observed spectrums.

## 4. Results

The analysis in this section focuses on evaluating the progress of electrical trees in epoxy resin using PRPD, PSA, and NLTSA techniques and comparing them to the results obtained using the PD waveform. The evolution of a variety of parameters derived from the waveform and others from the PRPD analysis is shown in Figure 3, calculated for 11 development intervals of the tree growth in Sample 50-B. The waveform parameters were $A_{t}$ and f, used previously with promising results in [15], and the parameters associated with PRPD analysis were $Q_{95}$, $P_{dis}$, and discharge rate n (#PDs/min). All these parameters were normalized by their maximum value to make the comparison between them and between samples easier. Additionally, Figure 3 shows the length of the tree, with vertical lines showing the most relevant events in its development: the time when the tree reaches a length of 30% ($T_{s}$) is marked with a dotted blue line, the crossing to the counter-electrode ($T_{c}$) is marked with a solid blue line, and the final breakdown ($T_{b}$) is marked with a dotted black line.

It can be observed that considerable changes occur in all the parameters in the crossing to the counter-electrode, which is coincident with the findings in [15].

Table 2 shows the percentage variations between pre-crossing (interval 8) and post-crossing (interval 10). The variation in the parameters was calculated using Equation (2), where P is any of the parameters; this procedure was used to directly compare the parameters and to make their visualization simple. Parameter $Q_{95}$ has a 40% variation between the pre-crossing and post-crossing stages, indicating the capacity to describe the stage of crossing from the tree to the counter-electrode. However, the $A_{t}$ parameter with a variation of 37% shows that the waveform also provides relevant information, which can be understood as complementary information. This shows that the two techniques can simultaneously provide relevant information for the identification of the cross.
(16)$$\Delta \left(\%\right)=\frac{P\left(10\right)-P\left(8\right)}{max\left\{P\left(1\right),\ldots ,\;P\left(10\right)\right\}}\cdot 100$$

### 4.1. Phase-Resolved Partial Discharge

Given the importance of crossing in the development of an electrical tree, it is relevant to visualize the changes that occur in PRPDs before and after the tree crosses the insulation gap. Figure 4 shows the PRPDs of intervals 8 and 10, visualizing variations in the generated patterns for pre- and post-crossing, respectively. In the case of pre-crossing (interval 8), the classic behavior of tree-type discharge is observed [30], and post-crossing (interval 10) presented a more atypical pattern with respect to past investigations, which could be distorted by mixed sources and phenomena, associated with the bridge that the tree has provided between the electrodes, i.e., between the energized needle and the grounded plane electrode. However, the pattern is similar in some respects to that observed during the development of a reverse tree in [31], where it is shown that these patterns are associated with the widening and advancement of a reverse tree which, in the case of Sample 50-B, could be associated with the development of dark branches. The similarities observed between the visualized PRPD patterns and those found in [31] are the widening of the phase window of PD occurrence and the increase in PD magnitude. In addition, two patterns are differentiated that could allude to the patterns of turtles and wings, with the turtles being the cumulus of discharges below 100 pC and the wings being the discharges above 100 pC with presence in a narrower phase band. In this context, two types of PD could be identified that differ from each other by variations in the charge conduction on the surface of the branches, as mentioned in [10].

In order to quantitatively visualize the variations in PRPD, the statistical moments associated with the skewness, kurtosis, mean value, and variance of the data should be compared with respect to the charge, positive cycle C(+), and negative cycle C(−), as shown in Figure 5, corresponding to the pre-crossing (interval 8) of Sample 50-B.

Figure 6 and Figure 7 compare the statistical moments obtained from the histograms of pre-crossing (interval 8) and post-crossing (interval 10). In Figure 6, evident variations are observed in the skewness (S) and kurtosis (K) of the charge histogram, showing that the discharges in the pre-crossing have a distribution with negative symmetry (S = −1.1) and of the mesokurtic type (K = 3.1), and in post-crossing, the discharges passed to a distribution with positive symmetry (S = 3.9) and leptokurtic type (K = 21.3). This indicates that the charge distributions present variations that allow electrical tree crossing identification. On the other hand, the variations in the phase histograms are not so evident, showing that the skewness of the positive and negative cycles tend towards a positive distribution of the data when the crossing occurs, and the kurtosis tends to go from a leptokurtic distribution to a mesokurtic distribution.

An increase in the mean value of charge can be seen in Figure 7, indicating that the discharges increase in magnitude in the post-crossing; the average value of the positive and negative cycle shows little variation, indicating that the discharges have the same phase of occurrence before and after tree crossing. On the other hand, the variance increases for both the charge and phase data, but the variation in the charge data is especially important, changing from approximately 16 pC^2^ to 2000 pC^2^, which in terms of the standard deviation is a variation from 4 pC to 44 pC, evidencing change in the order of magnitude.

Since the results shown in Figure 6 and Figure 7 are only for one test, it is not possible to draw conclusions regarding trends; therefore, Figure 8a,b present the results of statistical moments for the eight tests carried out at different frequencies and voltages. Figure 8a shows that the phase distributions present tendencies that are generally repeated in the positive and negative half-cycles. It is observed that the skewness has a distribution that tends to pass from positive to normal; however, the variation is small (within the range −1 to 1), indicating no clear changes. A similar situation occurs with the kurtosis and the mean value; in the case of kurtosis, the median changes from 3.5 to 3 for the negative cycle (C(−)) and from 4 to 3 for the positive cycle (C(+)), maintaining approximately mesokurtic distribution. On the other hand, the median of the mean value varies from 1.5 to 1.35 rad in C(+) and remains at 4.5 rad in C(−), showing that the mean value PD phase has little variation. Finally, the median of the variance is approximately double, evidencing that the data increase the dispersion in the post-crossing.

Figure 8b shows six graphs in logarithmic scale of the parameters of skewness, kurtosis, mean value, and variance, calculated for the charge magnitude histogram, showing that the variations in the parameters between the pre- and post-crossing are more evident than in the case of the phase variable shown in Figure 8a. It is observed that charge distribution becomes positively asymmetric and of the leptokurtic type when it crosses the insulation (i.e., it reaches the counter-electrode). It can also be observed that the average charge value increases by one order of magnitude and the variance increases by approximately two orders of magnitude; something similar occurs with $Q_{95}$ and $P_{dis}$, showing that discharges increase by magnitude and variability, with the latter being understood as a growth in the dispersion of the apparent charge data.

### 4.2. Pulse Sequence Analysis

The PSA patterns in the pre- and post-crossing intervals were analyzed to evaluate the ability of the PSA technique to identify the insulation crossing of the electrical tree and compare with the PRPD technique. Figure 9 shows the PSA plots formed before and after the tree crossing of Sample 50-B. The pre-crossing stage had the classic behavior of electrical treeing of six clusters (highlighted in red in the pre-crossing stage) seen in [23]. Clusters I, II, and III are associated with discharges of the positive cycle and clusters IV, V, and VI with discharges of the negative cycle. In addition, two clusters (highlighted in blue) that appear towards the end of the pre-crossing interval are associated with noise. On the other hand, in post-crossing stage, the six clusters were also observed (highlighted in red), but also, four clusters (highlighted in green) appeared when the tree crossing occurred which could be associated with new discharges. There is also a cluster of discharges on the axis of the line x = y (highlighted in yellow), associated with discharges with a higher apparent charge (above 50 pC), similar to the pattern of discharges that has been observed close to breakdown in [23].

Figure 10 shows the PSA formed for the pre- and post-crossing for Samples 50-A, 350-B, and 550-B. In the three pre-crossing patterns, the six clusters already defined and highlighted in red in Figure 9 could be identified. On the other hand, in the pre-crossing stage of Samples 50-A and 350-B, patterns similar to the clusters highlighted in green in Figure 9 are identified, and in the Sample 550-B, the pattern of the clusters highlighted in blue can be observed. When analyzing the patterns formed in the post-crossing stage, it can be seen that in Sample 550-B, the six clusters are clearly repeated, and in Samples 50-A and 350-B a pattern that only identifies four clusters was observed. Additionally, the pattern identified with green circles can only be identified in Sample 550-B. This indicates that the formation of patterns with the same characteristics of the ones highlighted in red, green, or blue in Figure 9 would not be identifiers of tree crossing, since they appear in the PSA patterns of the pre- or post-crossing stages indistinctly. Finally, the PSA patterns analyzed from the post-crossing systematically show the pattern located in the line x = y (in yellow in Figure 9) associated with large discharges, indicating that it would serve as an evident identifier of the crossing or the development of dark branches, which occurs right after crossing [10]. This result would have a comparative advantage over the rest of the PRPD and waveform parameters, since such the PSA pattern could be considered the only pattern that would not need the PD measurement in previous stages in order to identify the post-crossing stage.

### 4.3. Nonlinear Time Series Analysis

Figure 11 summarizes the correlation dimension result for all the samples at pre-crossing and post-crossing. As this box plot shows, on average, the correlation dimension decreases after reaching the counter electrode, which agrees with the result observed in [32]. This decrease may indicate a less complex dynamic after the electrical trees cross the insulation [13]. Figure 11 also shows that the dispersion in the correlation dimension for the post-crossing stage is greater than in the pre-crossing. The higher dispersion observed in the values of this parameter may limit its utility for diagnostic purposes.

### 4.4. Partial Discharge Waveform Parameters

Figure 12 shows the average values of the waveform parameters calculated for the eight samples and for the pre- and post-crossing stages. The trends already found in [15] are observed, providing complementary information to the results of the PRPD parameters and PSA plots. Parameters $A_{t}$, PRL, and $\sigma_{T}$ tend to decrease towards post-crossing, confirming that the PDs have a relatively shorter equivalent time compared to pre-crossing, indicating a shorter duration of discharges. The most evident variation is shown by the parameter $A_{t}$, which represents the area of the pulse tail, decreasing from a median close to 100 ns to a median of approximately 80 ns. On the other hand, it is observed that the parameters f, PRH, and $\sigma_{F}$ increase towards the post-crossing, again indicating the presence of shorter pulses; that is, the frequency content moves towards frequencies greater than the order of 50–60 MHz. 

In general, Figure 12 shows a decrease in the box size due to a decrease in the variability in all parameters between samples, showing similar behavior of the PD in the post-crossing stage in all samples. This could mean that the waveform parameters are not influenced by the characteristics of the input signal, but mainly by the condition of the local electric field and the structure of the tree. However, this assertion must be verified with further experiments and analyses. The variability shown in the pre-crossing stage can be associated with the fact that PD measurements prior to crossing are difficult to perform, since highly sensitive equipment is required to detect the small pulses usually characteristic of this stage of development. On the other hand, in the post-crossing stage, there is greater PD activity, which favors the visualization of trends.

## 5. Discussion

Figure 13 shows a comparison of the parameters obtained from PRPD analysis that present the best behavior to describe or identify tree crossing. It is observed that the data are in a wide range of values (two or three orders of magnitude), indicating that the parameters calculated directly from the PRPD patterns could be sensitive to variations in the shape of the PRPD generated by the presence of noise or discharges from other sources not detected with the naked eye. On the other hand, Figure 13 shows the comparison of the calculated parameters of the waveform, showing that the variability between the averages of the parameters of each sample decreases in comparison with the PRPD parameters, becoming evident in the post-crossing stage, where it is observed that practically, the eight samples analyzed show values close to each other in all the parameters, proving to be more robust to variations in the applied frequency and voltage. This may be due to the fact that, by having the waveform of the discharges, it is possible to separate the discharges from other sources by means of algorithms and, above all, separate the noise, a procedure that is explained in Section 2 and discussed in more detail in ref. [18]. Additionally, since the analysis is made on each pulse individually and not the distribution of the data, as in PRPD, it is not sensitive to errors in the phase measurement or in the sequence of the pulses; that is to say, the result would not substantially change in the case of a phase shift or if, for some reason, the measurement system skips several points in the sequence.

The previous analysis shows that the parameters calculated from the PRPDs could present problems as predictors of the development stages of trees due to their high variability between samples; so, considering the possibility of applying machine learning to identifying stages of development could complicate its application. However, this would not occur with the waveform parameters, since it was observed that they present more stable and robust behavior when dealing with changes in the applied voltage and frequency, especially in the post-crossing stage.

## 6. Conclusions

We compared techniques for characterizing electrical tree growth in epoxy resin to evaluate their ability to identify tree bulk-insulation crossing, a key event that is a precursor to the catastrophic failure of high-voltage solid polymeric insulation. We deployed traditional techniques such as phase-resolved partial discharge (PRPD) and pulse sequence analysis (PSA), and more recent techniques such as the tools of nonlinear time series analysis (NLTSA) and PD pulse waveform analysis. 

Both traditional partial discharge analysis techniques (PRPD and PSA) had the ability to identify the crossing of the electrical tree to the counter-electrode. The PRPD analysis showed sensitivity in the charge histogram parameters, displaying a distribution that tends to positive asymmetry and the leptokurtic type in the post-crossing; the parameters of mean value ($Vm_{q}$), variance ($Var_{q}$), percentile 95 ($Q_{95}$), and dissipated power ($P_{dis}$) increased their value by at least one order of magnitude when crossing the counter-electrode. The PSA graphs showed discharge patterns that were along the line x=y, observed exclusively in the post-crossing stage; the pre-crossing stage was characterized by presenting graphs of six well-defined clusters, which are deformed in the post-crossing stage, with only four being distinguished. The characteristic of the nonlinear dynamic system evaluated here was the correlation dimension. It was found that the correlation dimension decreased after the tree reached the counter electrode, indicating that the system changes to a less complex dynamic after crossing. However, the complexity and demanding calculation of the method makes it less applicable than the others.

The waveform parameters show variation in the tree bulk-insulation crossing event, thus identifying the moment when this stage is reached. In general, PDs tend to be shorter and faster pulses in post-crossing; that is, the temporal parameters ($A_{t}$ and $\sigma_{T}$) decrease and the frequency parameters (f and $\sigma_{F}$) increase, showing a movement of spectral power towards the high frequency range (PRL decreases and PRH increases). On the other hand, it was observed that the PD waveform parameters presented less variability between samples, making them more robust to variations in the applied voltage and frequency, and so to use in a broader range of testing situations. Moreover, this method presents an advantage over traditional techniques when it comes to automating the recognition of stages using artificial intelligence and machine learning. 

Overall, it is concluded that techniques based on the information that the PD pulse waveform provides are the most appropriate methods for assessing the aging state of epoxy resins due to electrical tree propagation. While the focus of this study was on evaluating the ability of these techniques to identify tree insulation crossing, future research can expand on these findings to characterize the type and propagation of electrical trees. By developing more advanced criteria and intelligent systems for insulation diagnosis, researchers can enhance the accuracy and efficiency of the analysis and gain a more detailed understanding of the aging process. The insights gained from this study have the potential to inform more effective maintenance and replacement strategies, thus contributing to the development of more reliable and sustainable power systems.

## Figures and Tables

**Figure 1 polymers-15-02461-f001:**
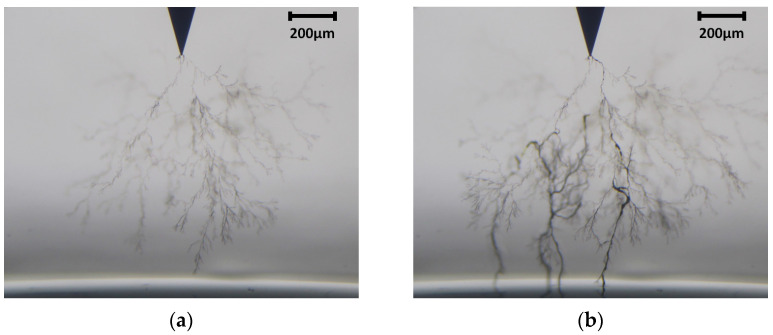
Optical image of electrical tree of Sample 50-B. (**a**) Before pre-crossing; (**b**) before breakdown.

**Figure 2 polymers-15-02461-f002:**
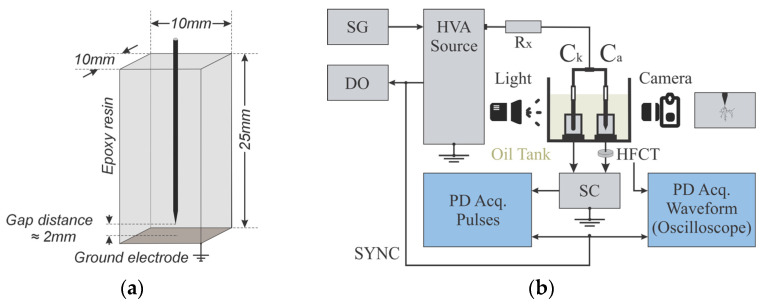
(**a**) Test sample made of epoxy resin. (**b**) Testing circuit for electrical treeing.

**Figure 3 polymers-15-02461-f003:**
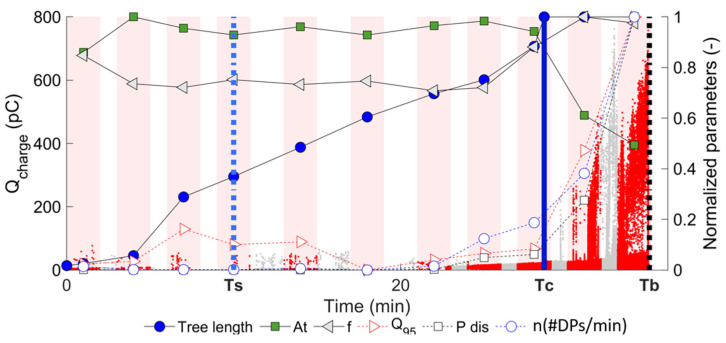
Evolution of PRPD and waveform parameters for Sample 50-B.

**Figure 4 polymers-15-02461-f004:**
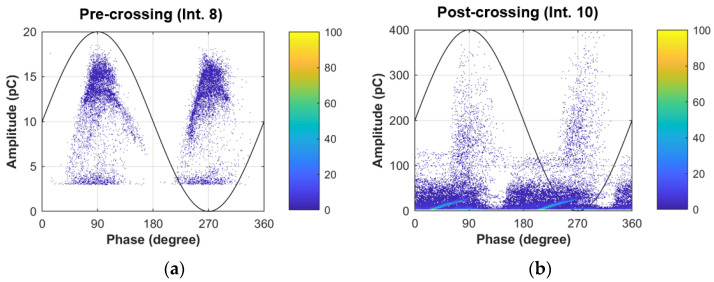
PRPD plots for: (**a**) pre-crossing (Int. 8) and (**b**) post-crossing (Int. 10) of Sample 50-B.

**Figure 5 polymers-15-02461-f005:**
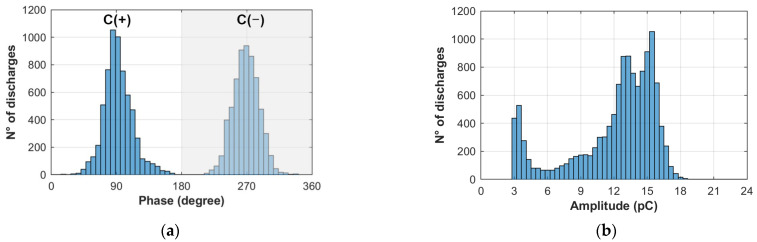
Histograms of (**a**) phase and (**b**) amplitude for the PRPD of pre-crossing (Int. 8) Sample 50-B.

**Figure 6 polymers-15-02461-f006:**
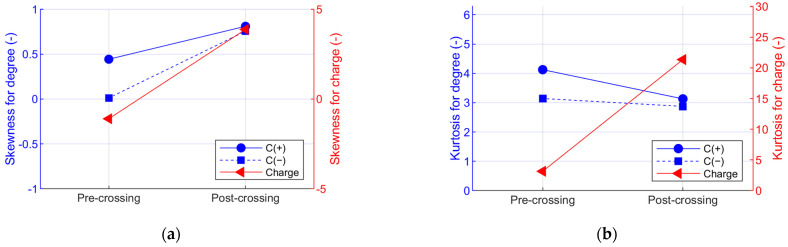
(**a**) Skewness and (**b**) kurtosis parameters calculated for pre-crossing (Int. 8) and post-crossing (Int. 10) of Sample 50-B.

**Figure 7 polymers-15-02461-f007:**
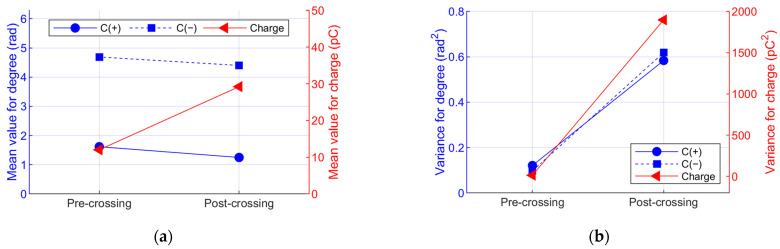
(**a**) Mean value and (**b**) variance parameters of the phase and charge pre-crossing (Int. 8) and post-crossing (Int. 10) of Sample 50-B.

**Figure 8 polymers-15-02461-f008:**
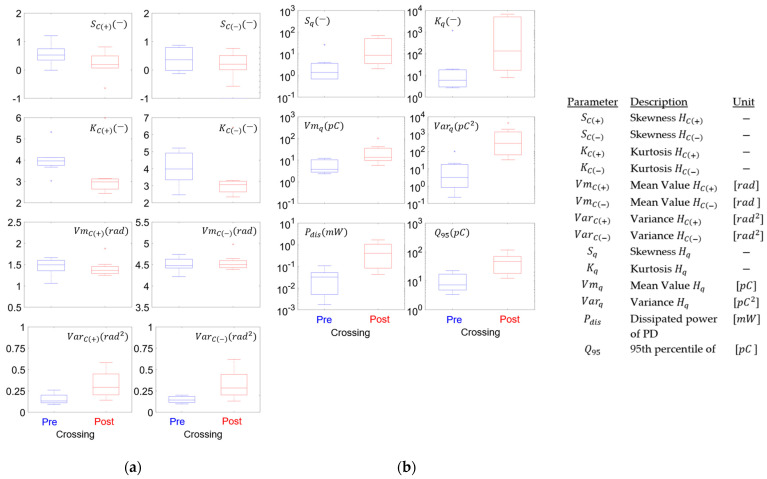
Parameters in PRPD for eight samples in pre-crossing and post-crossing (C(+): positive cycle; C(−): negative cycle). (**a**) Statistical moments of phase; (**b**) statistical moments of charge magnitude, *P_dis_* and *Q*_95_.

**Figure 9 polymers-15-02461-f009:**
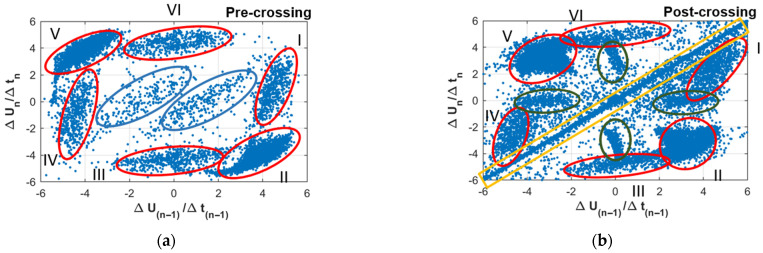
PSA plot for (**a**) pre-crossing (Int. 8) and (**b**) post-crossing (Int. 10) of Sample 50-B. Clusters highlighted in red (I–VI) showing classic behavior of electrical treeing, clusters highlighted in green that appeared after tree-crossing, and cluster highlighted in yellow rectangle associated with discharges of higher apparent charge evidencing closeness to breakdown.

**Figure 10 polymers-15-02461-f010:**
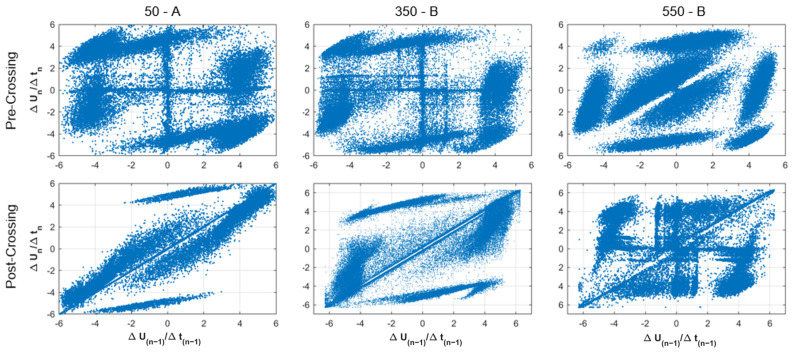
PSA plots of pre and post-crossing intervals for Samples 50-A, 350-B, and 550-B. Top row: pre-crossing; bottom row: post-crossing.

**Figure 11 polymers-15-02461-f011:**
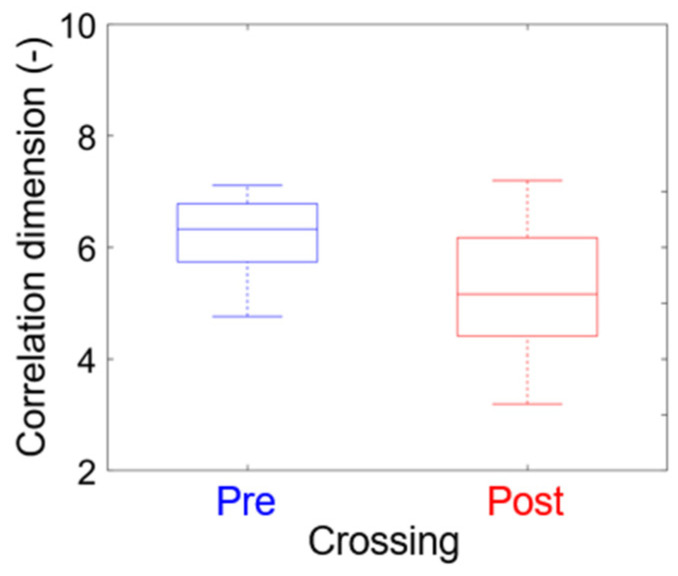
Correlation dimension of pre and post-crossing for all eight samples.

**Figure 12 polymers-15-02461-f012:**
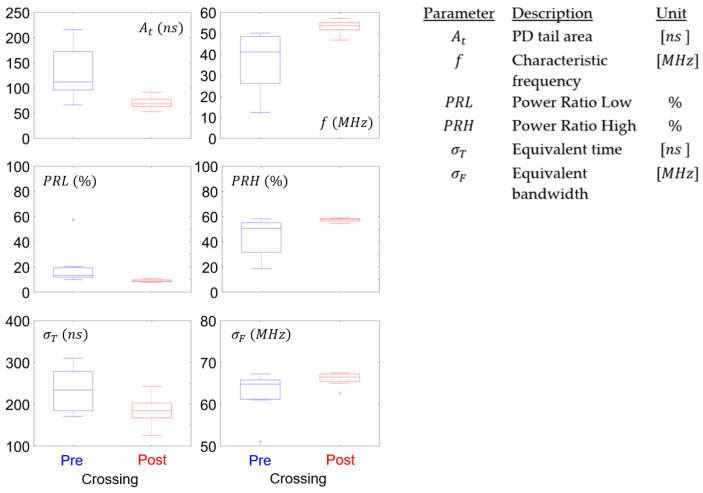
Waveform parameter results for all eight samples.

**Figure 13 polymers-15-02461-f013:**
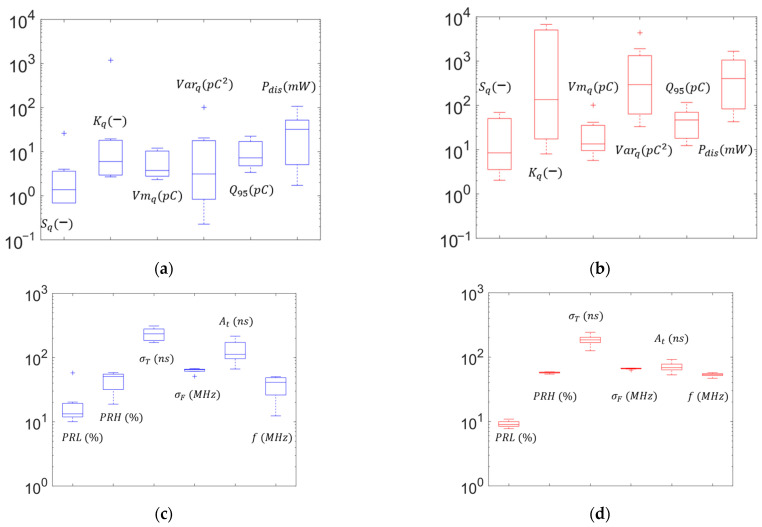
(**a**,**b**) PRPD charge parameters, (**c**,**d**) waveform parameters of the eight samples in pre-crossing (**a**,**c**) and post-crossing (**b**,**d**) stages.

**Table 1 polymers-15-02461-t001:** Test specimens for electrical tree growth and their testing conditions.

Sample	Voltage (kV)	Frequency (Hz)
50-A	10	50
50-B	14	50
150-A	10	150
150-B	14	150
350-A	10	350
350-B	12	350
550-A	10	550
550-B	14	550

**Table 2 polymers-15-02461-t002:** Percentage variation between the parameters of pre-crossing (interval 8) and post-crossing (interval 10) of Sample 50-B.

Parameter	Interval 8	Interval 10	Δ (%)
$A_{t}$ (ns)	113.4	70.5	−37.2
f (MHz)	39.3	54.5	+27.9
$Q_{95}$ (pC)	16.21	116.4	+40.7
$P_{\mathrm{dis}}$ (mW)	0.02	0.112	+22.7
n (#PDs/min)	6284.5	19362	+25.8

## Data Availability

The data presented in this study are available on request from the corresponding author.

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
