# Peer review of "Identification of Electrical Tree Aging State in Epoxy Resin Using Partial Discharge Waveforms Compared to Traditional Analysis"

_polymers, 2023, doi:10.3390/polym15112461_

Round 1

Reviewer 1 Report

This study compares different partial discharge (PD) analysis techniques to identify the crossing of electrical trees in epoxy resin, which is a key event that precedes the failure of high voltage solid polymeric insulation. The study deploys both traditional techniques, such as phase-resolved PD (PRPD) and pulse sequence analysis (PSA), and more recent techniques, such as nonlinear time series analysis (NLTSA) and PD pulse waveform analysis. The study finds that both PRPD and PSA can identify the tree crossing event, with PRPD showing sensitivity in the charge histogram parameters, and PSA showing discharge patterns that are along the line x=y in the post-crossing stage. NLTSA, however, is less applicable due to its complexity and calculation demands. PD waveform parameters, on the other hand, show variation in the tree bulk-insulation crossing event, making them more appropriate for evaluating the ageing state of epoxy resins due to electrical tree propagation. This method is also more robust to variations in the applied voltage and frequency and can be used in a broader range of testing situations. The study recommends developing more criteria and intelligent systems for insulation diagnosis based on PD waveform analysis. Overall, the study provides valuable insights into the characterization of electrical tree growth in epoxy resin and can inform the development of more effective diagnostic tools for asset management of high voltage polymeric insulation.

 As an expert in the field, I think the study is well-designed and executed, and the results are well-presented and analyzed. The study makes a significant contribution to the understanding of electrical tree growth in epoxy resin and can inform the development of more effective diagnostic tools for asset management of high voltage polymeric insulation. The study's conclusion that PD waveform analysis is the most appropriate method for evaluating the ageing state of epoxy resins due to electrical tree propagation is convincing, and the recommendation to develop more criteria and intelligent systems for insulation diagnosis is appropriate. I recommend the paper for publication with minor revisions.

The authors should clarify some technical terms and provide more context and background information for readers who are not experts in the field. For example, they should provide more information on the partial discharge (PD) analysis techniques used, such as PRPD and PSA, and explain how they work and what they measure. They should also explain the significance of the correlation dimension in NLTSA and provide more context on PD waveform parameters. Additionally, they should provide more information on the limitations and challenges of the different techniques and discuss how they can be addressed in future research.

Finally, the authors should provide more details on the materials and methods used in the study, including the specific equipment and settings used for the PD measurement systems and analysis techniques. They should also provide more information on the sample size and variability between samples and discuss the potential impact on the study's findings. Overall, the paper is well-written and provides valuable insights into the characterization of electrical tree growth in epoxy resin, and I recommend it for publication with minor revisions.

This study compares different partial discharge (PD) analysis techniques to identify the crossing of electrical trees in epoxy resin, which is a key event that precedes the failure of high voltage solid polymeric insulation. The study deploys both traditional techniques, such as phase-resolved PD (PRPD) and pulse sequence analysis (PSA), and more recent techniques, such as nonlinear time series analysis (NLTSA) and PD pulse waveform analysis. The study finds that both PRPD and PSA can identify the tree crossing event, with PRPD showing sensitivity in the charge histogram parameters, and PSA showing discharge patterns that are along the line x=y in the post-crossing stage. NLTSA, however, is less applicable due to its complexity and calculation demands. PD waveform parameters, on the other hand, show variation in the tree bulk-insulation crossing event, making them more appropriate for evaluating the ageing state of epoxy resins due to electrical tree propagation. This method is also more robust to variations in the applied voltage and frequency and can be used in a broader range of testing situations. The study recommends developing more criteria and intelligent systems for insulation diagnosis based on PD waveform analysis. Overall, the study provides valuable insights into the characterization of electrical tree growth in epoxy resin and can inform the development of more effective diagnostic tools for asset management of high voltage polymeric insulation.

 As an expert in the field, I think the study is well-designed and executed, and the results are well-presented and analyzed. The study makes a significant contribution to the understanding of electrical tree growth in epoxy resin and can inform the development of more effective diagnostic tools for asset management of high voltage polymeric insulation. The study's conclusion that PD waveform analysis is the most appropriate method for evaluating the ageing state of epoxy resins due to electrical tree propagation is convincing, and the recommendation to develop more criteria and intelligent systems for insulation diagnosis is appropriate. I recommend the paper for publication with minor revisions.

The authors should clarify some technical terms and provide more context and background information for readers who are not experts in the field. For example, they should provide more information on the partial discharge (PD) analysis techniques used, such as PRPD and PSA, and explain how they work and what they measure. They should also explain the significance of the correlation dimension in NLTSA and provide more context on PD waveform parameters. Additionally, they should provide more information on the limitations and challenges of the different techniques and discuss how they can be addressed in future research.

Finally, the authors should provide more details on the materials and methods used in the study, including the specific equipment and settings used for the PD measurement systems and analysis techniques. They should also provide more information on the sample size and variability between samples and discuss the potential impact on the study's findings. Overall, the paper is well-written and provides valuable insights into the characterization of electrical tree growth in epoxy resin, and I recommend it for publication with minor revisions.

Reviewer 2 Report

I would like to thank the authors for a valuable manuscript, which deals with a very interesting and current topic. The manuscript deals with the problem of identifying electrical treeing in epoxy resin and provides information on several available analyses. Overall, the paper is well designed and shows, with links to other literature, that the topic is being intensively studied by the team of authors. However, at least at the end of the manuscript, I would have appreciated a note on whether it is possible to characterize the type of tree and its propagation in the material based on the analyses. I recommend the manuscript for publication after solving the comments below. 

Major comments 

On the line 178 you state "Specifically, the parameters ??, ?, ??, ??, PRL and PRH were used" - Of course it is possible to refer to other articles and e.g. the references in the article to the circuit diagram etc. are correct, but at least the parameters that are evaluated should be described in the article and used abbreviations and symbols should be explained. I should know directly from the article what is measured and evaluated - e.g. what the parameter "discharge rate PDs/min" means and what is the unit of this parameter (dimensionless according to the graph or does it have a primary unit).

On the line 378 in the Discussion section you state "procedure that was explained in ref. [29]" - I would expand on this information in the Discussion section, so that the reader of this manuscript can understand at least the basic idea of the procedure. Earlier in the manuscript, on lines 249-250, you state "This indicates that the charge distributions present variations that allow electrical tree crossing identification.", and I lack more specific information about whether it is possible to detect the tree crossing or whether it is possible to characterize the tree more precisely on the basis of performed analyses.

In the list of references, you cite the same document as source 18 and 29 - please carefully review the list of used references and remove any duplications and inconsistencies. 

I counted more than one-third of the papers in the reference list in which someone from the author collective of the manuscript is the author. It is certainly possible to use this higher number of citations for your own articles if they are relevant to the topic and allow you to optimally broaden the overview of the topic, but I would expect more articles from other authors - more to compare the results with those of other research teams. 

Figure 7 summarises the results of the statistical analysis, but may be a bit confusing for the less informed reader. In terms of graphic design, I would use colour to at least distinguish the Pre and Post phases. The image could also be enlarged a little more. Overall, I would have included in the manuscript, for example, a table summarizing all parameters presented with a symbol and unit that the reader could quickly understand. I would add the same graphical layout and summarization of the presented parameters in the case of Figure 11.

You present in Figures 12 and 13 the values of a number of monitored parameters and connect different parameters with a line. From the point of view of correct formal notation, the different parameters should only be shown as separate points. From the point of view of the nature of the data, bar charts would be better, but I understand that they would be unclear in terms of the volume of displayed data, so it is possible to stay with a scatter plot.

Minor comments

I would mention Electrical treeing is... in text "Electrical trees are one of the main degradation mechanisms" on line 13.

On line 45 you list "transformers, dry type transformers..." - I would list only one - general or specific.

Please also list the manufacturer for Mepox 1124(cl)/2262 resin.

The titles of subsections 3.1, 3.2 and 3.3, as well as 4.1, 4.2 4.3, should be named with the full title and the abbreviation used only in the text.

Figure 2 is important in evaluating the results from the measurements, but I evaluate it as relatively not clear. I would at least expand the figure to the width of the full page and better assign the displayed curves to the y-axes - is the right y-axis (Normalized parameters) assigned to all monitored parameters?

On line 240 you write "shown0020in Figure 4" - I assume this is a typing error.

On line 248 you write "(K=3.)" - please edit the notation of the number I assume with the missing number of decimal places.

Round 2

Reviewer 2 Report

I thank the authors for editing the manuscript. In the current version, I find the article more clear, better organized as a complete work, and better characterized in the performed analyses. I have no further major comments on the manuscript in the current version and I would only be able to discuss minor issues related to the layout and structuring of the text, but these are quite subjective. In my opinion, the paper provides everything that the authors wanted to present to the scientific community.